# Respiratory symptoms and respiratory deaths: A multi-cohort study with 45 years observation time

Knut Stavem[1,2,3]*, Ane Johannessen[4], Rune Nielsen[5], Amund Gulsvik[5]

**1** Pulmonary Department, Akershus University Hospital, Lørenskog, Norway, **2** Institute of Clinical Medicine, University of Oslo, Oslo, Norway, **3** Health Services Research Unit, Akershus University Hospital, Lørenskog, Norway, **4** Centre for International Health, Department of Global Public Health and Primary Care, University of Bergen, Bergen, Norway, **5** Department of Clinical Science, Faculty of Medicine, University of Bergen, Bergen, Norway

* knut.stavem@medisin.uio.no

**Data Availability Statement:** Our ethical approval granted by the Regional Committees for Medical and Health Research Ethics in Norway does not allow public sharing of the data. A data set can be

## Abstract

This study determined the association between respiratory symptoms and death from respiratory causes over a period of 45 years. In four cohorts of random samples of Norwegian populations with 103,881 participants, 43,731 persons had died per 31 December 2016. In total, 5,949 (14%) had died from respiratory diseases; 2,442 (41%) from lung cancer, 1,717 (29%) chronic obstructive pulmonary disease (COPD), 1,348 (23%) pneumonia, 119 (2%) asthma, 147 (2%) interstitial lung disease and 176 (3%) other pulmonary diseases. Compared with persons without respiratory symptoms the multivariable adjusted hazard ratio (HR) for lung cancer deaths increased with score of breathlessness on effort and cough and phlegm, being 2.6 (95% CI 2.1–3.2) for breathlessness score 3 and 2.1 (95% CI 1.7–2.5) for cough and phlegm score 5. The HR of COPD death was 6.4 (95% CI 5.4–7.7) for breathlessness score 3 and 3.0 (2.4–3.6) for cough and phlegm score 5. Attacks of breathlessness and wheeze score 2 had a HR of 1.6 (1.4–1.9) for COPD death. The risk of pneumonia deaths increased also with higher breathlessness on effort score, but not with higher cough and phlegm score, except for score 2 with HR 1.5 (1.2–1.8). In this study with >2.4 million person-years at risk, a positive association was observed between scores of respiratory symptoms and deaths due to COPD and lung cancer. Respiratory symptoms are thus important risk factors, which should be followed thoroughly by health care practitioners for the benefit of public health.

## Introduction

Respiratory diseases, such as pneumonia, lung cancer and chronic obstructive pulmonary disease (COPD), remain a leading cause of disability and death worldwide [1–3]. Respiratory symptoms are important indicators of these diseases, and recording of respiratory symptoms is a cheap and easy screening method.

In the 1970s, at least one respiratory symptom was reported by 40% of the adult inhabitants of the Oslo city: cough in the morning, attacks of breathlessness, breathless when climbing

made available for scientific analysis on request, provided that the respective research institution proofs handling of the data strictly in accordance with ethical regulations (written ethics protocol, full compliance with the Declaration of Helsinki). To ensure full anonymity only the main variables of the final analyses are provided. We confirm that the data file provided constitutes the minimal data set necessary to replicate the findings of your study in their entirety. Data requests can be made to corresponding author Knut Stavem (knut. stavem@medisin.uio.no<mailto:knut. stavem@medisin.uio.no>) Contact information to the Regional Ethics Committee: REC West - Secretariat University of Bergen, Faculty of Medicine, P.O. Box 7804, 5020 Bergen email: post@helseforskning.etikkom.no.

**Funding:** The authors received no specific funding for this work.

**Competing interests:** I have read the journal's policy and the authors of this manuscript have the following competing interests: RN reports grants and personal fees from AstraZeneca, grants from GlaxoSmithKline, grants from Boehringer Ingelheim, outside the submitted work. All other authors have declared that no competing interests exist. This does not alter our adherence to PLOS ONE policies on sharing data and materials.

stairs or wheezing [4]. The symptom burden remains high also in recent times, with approximately 20% reporting wheeze and >25% cough [5].

Respiratory symptoms are associated with all-cause and cardiovascular mortality [6–9]. Less is known about the association between respiratory symptoms and respiratory causes of death. The limited number of respiratory deaths in previous studies [10–12], and the non-random nature of some of the cohorts [10], limit generalisability of findings. In a recent general population study with <15 years of follow-up time, individuals with chronic respiratory symptoms and normal spirometry had increased mortality due to respiratory diseases [8, 13].

Large cohorts are needed to observe a sufficient number of deaths due to asthma and interstitial pneumonia, which are rare causes of death in the general population. We therefore extended a previous analysis [12] to investigate the long-term relationship between respiratory symptoms and mortality for several respiratory diseases by combining four general population cohorts [6].

This study reports on the association of self-reported subsets of respiratory symptoms such as breathlessness on effort, cough and phlegm, and attacks of breathlessness/wheeze with the mortality from lung cancer, pneumonia/influenza, COPD, asthma and interstitial pneumonias over 45 years.

## Materials and methods

### Study population

This multicohort study used harmonized individual-level data from four cross-sectional surveys in the City of Oslo 1972 and 1998–1999, Hordaland County in 1985, 1988–1989 (including Sauda municipality in Rogaland county) and 1998–1999 [14]. Oslo, the capital of Norway, had 477,476 inhabitants in 1972 and 499,693 persons in 1998. Hordaland county is a combined rural and urban (Bergen) population with 399,702 persons in 1985, 405,063 in 1988 and 428,823 in 1998. Sauda is a rural municipality with 5,416 inhabitants in 1988.

The target populations were born 1902–1973. The sample frames were updated lists from the Norwegian National Population Registry. Invitees were drawn at random for the 1972 (Oslo 72), 1985 (Hordaland 85) and 1998–1999 (Oslo/Hordaland 98–99) surveys. The 1988–1990 survey (Støvlunge 88–90) invited all men born 1914–1958, plus a 10% sample of the general population of City of Bergen examined in a previous cohort in 1965–1970 [14, 15], leading to about 6% women in this cohort. We excluded those included in one cohort from later cohorts. The eligible population sample comprised 158,702 unique persons. We have previously presented the recruitment and pooling of these cohorts [6].

The analyses included respondents who provided information on smoking, education, occupational exposure to dust/gas and respiratory symptoms, in total 103,881 persons; 65% of the sample of unique persons (Table 1).

The study was approved by the Committee on Medical Research Ethics (reference 2017/ 1679), The Norwegian Data Inspectorate (07/00414) and The Norwegian Directorate of Health (07/948).

### Questionnaire

We used a questionnaire, which was a modification of one approved by the British Medical Research Council's (MRC) Committee on Research into Chronic Bronchitis in 1966. The validity of the Norwegian respiratory questionnaire has been evaluated [16], and compared with the original MRC questionnaire [17].

The questionnaire included 11 questions about respiratory symptoms (S1 Table), covering current cough, phlegm, wheezing, periods of cough and/or phlegm and breathlessness, which

**Table 1. Flowchart of randomly sampled individuals in the study of respiratory symptoms and all cause deaths in Norway.**

| | Oslo cohort 1972 | Hordaland county cohort 1985 | Hordaland county and Sauda municipality Rogaland county 1988–1990 | Hordaland and Oslo counties cohort 1998–1999 | Total |
|---|---|---|---|---|---|
| Sample drawn from target populations | 19998 | 4992 | 112235 | 25000 | 162225 |
| Present in previous samples, excluded | 1* | 2 | 1893 | 1627 | 3523 |
| Eligible, unique persons | 19997 | 4990 | 110342 | 23373 | 158702 |
| Eligible after missing times excluded | 19892 | 4982 | 108812 | 23210 | 156896 |
| Respondents (response to at least 1 of 17 questionnaire items) | 17690 | 4461 | 77003 | 15870 | 115024 |
| Respondents to smoking habits | 17680 | 4404 | 76675 | 15623 | 114380 |
| Respondents to smoking habits and education | 17377 | 4347 | 75406 | 14994 | 112124 |
| Respondents to smoking habits, education and occupational exposure | 16445 | 4307 | 71958 | 14765 | 107475 |
| Respondents to all items on respiratory symptoms (in analysis) | 16084 | 4137 | 69168 | 14492 | 103881 |

*removed duplicate record.

were aggregated into three symptom groups: breathlessness scored 0 to 4; cough and/or phlegm symptoms scored 0 to 5, and attacks of breathlessness and wheezing scored 0 to 2 [6]. A higher score represents more severe symptoms.

The questionnaire also included questions on smoking history [18] and occupational exposure to air pollution. Smoking was categorized as current smokers (daily at the time of the study), ex-smokers, or never-smokers. Tobacco consumption was estimated from number of cigarettes per day (1 cigarette = 1 g), and grouped as <10 g, 10–19 g and ≥20 g. Occupational exposure to air pollution was defined by responding "yes" to "Have you been exposed to particles, gases or damp at your working place?"

No self-reported cardiopulmonary disease was defined as negative replies to 10 questions on treatments by physician or hospitals of asthma, bronchitis, emphysema, pleuritis, lung tuberculosis, other pulmonary disease, myocardial infarction, angina pectoris, hearth failure or other hearth diseases.

## Follow-up and census data

Date of death, emigration and cause of death until 31 December 2016 were obtained from the National Cause of Death Registry. All inhabitants of Norway have a unique personal identification number that allows complete follow-up until death or emigration. In total, 156,896 persons were initially observed. A total of 103,881 respondents with known smoking status and complete responses were followed; median follow-up was 27.4 years, maximum 45.2 years. They represented 2,449,538 person-years at risk.

Highest attained education was extracted from the national census for each decade and grouped according to the maximum length of education using three levels: compulsory education (7–10 years), medium level (11–13 years) and university level (≥14 years).

The overall inclusion rate in the analyses was 65% of the target population and varied for the baseline examination from 81% of the target population of the cohort of 1972 to 62% in the cohort of 1998–1999. The distribution of participants of the various groups was almost identical in the target population and the population of respondents with regard to age, sex and education (S2 Table).

**Table 2. Classification of underlying causes of death.**

| Respiratory causes | EU 2012 classification | ICD-8 | ICD-9 | ICD-10 |
|---|---|---|---|---|
| Lung cancer (trachea, bronchus, lung) | 15 | 162.0–162.9 | 162.0–162.9 | C33, C34.0–C.34.9 |
| COPD | 56 | 491–492, 518 | 491–492, 494, 496 | J42, J43.9, J44.0-J44.9, J47 |
| Pneumonia incl. influenza | 52, 53 | 480–480, 470–474 | 481,482, 485–486, 470, 471, 487 | J10.1, J11.0-J11.1, J13–J14, J15.2–J15.9, J18.0–J18.9 |
| Asthma | 55 | 493 | 493, 493.9 | J45.0, J45.9, J46 |
| Interstitial pneumonias* | 57 (subset) | 515, 516.3, 516.9 | 515, 516,3, 516.9 | J84.0, J84.1, J84.9 |
| Other respiratory | 57 (excl. interstitial pneumonias*) | 460–478, 495, 500–519 (excl.*) | 460–478, 495, 500–519 (excl. *) | J00-J06, J20–J39, J60–J99 (excl. *) |

## Classification of causes of death

Classification of respiratory causes as underlying causes of death was done using the European Shortlist for Causes of Death, 2012 [19, 20], with two modifications: 1) Because there were few cases with influenza, this was combined with pneumonia; 2) Interstitial pneumonias were extracted from "Other diseases of the respiratory system" and analyzed as a separate entity (Table 2).

## Statistical analysis

Descriptive statistics are presented in frequency tables as number (%). For start dates for observations in the cohorts, we used 5 October 1972 in the Oslo county 1972 cohort, the 15th of the actual starting month in the Hordaland county and Sauda municipality cohort 1988–1990, and the actual start date in the other two cohorts. Other missing start dates for participants were imputed using the median start date in the same cohort: 31 December 1989 in the Hordaland county and Sauda municipality cohort 1988–1990 (n = 100) and 5 October 1998 in the Oslo and Hordaland counties cohort 1998–1999 (n = 4).

The cohort members were followed until death or censored at the date of emigration or end of follow-up on 31 December 2016, whichever came first. For some people that emigrated (n = 231), we did not have a date of emigration, but only an interval. These cases were censored at the mid-point of the interval [6]. We did not impute missing values for other variables.

We pooled the four cohorts and analyzed the association of symptom scores of breathlessness on effort, cough or/and phlegm and attacks of breathlessness or/and wheeze with cause-specific mortality. We used Cox proportional hazards analysis, with age as the dependent variable [21]. We also repeated the analysis replacing the respiratory symptom scores with a variable with no respiratory symptom = 0, any respiratory symptom = 1. These analyses were prepared using shared frailty for study cohort, i.e. incorporating cluster-specific random effects to account for within-cluster homogeneity in outcomes [22].

All analyses were multivariable, adjusting for sex, education (<10, 11–13, ≥14 years), smoking habits (never, ex-, current-smoker) and occupational exposure (dust/fume vs. none). The results are presented as hazard ratios (HR) of death with 95% confidence intervals (CI).

Finally, we conducted analyses in strata of the pooled sample (men, women, never smokers, those without a history of cardiopulmonary disease) according to death from lung cancer, COPD and pneumonia using the same approach and the same covariates. For other causes of death, we did not conduct stratified analyses because of few events.

The proportional hazards assumption was checked graphically using log-log plots and was considered acceptable. We chose a significance level of 0.05 using two-sided tests. Stata version 16.1 (StataCorp, College Station, TX, USA) was used for all statistical analyses.

## Results

Among the 103,881 individuals (response rate 64%), 78% were men, and mean baseline age was 46.8 years (range 15–92 years). Altogether 43,731 (42%) had died as of 31 December 2016 and had a specified cause of death. In total 5,949 persons (14% of all deaths) had a pulmonary death; lung cancer 2,442 (41%), COPD 1,717 (28%), and pneumonia 1348 (23%) including 40 cases of influenza deaths (Table 2). In addition, 147 (2.5%) died from interstitial pneumonia, 119 from asthma (2.0%) and 176 due to other pulmonary causes (3.0%).

The prevalence of baseline symptom scores and the distribution of symptoms according to the principal causes of death is shown in S3 Table.

### Respiratory mortality

The overall crude mortality rate (MR) of respiratory diseases was 243 per 100,000 person-years. In total, 6.1% of men (4971/81510) and 4.4% of women (978/22371) died from respiratory causes. Among individuals with pulmonary deaths, 87% were ever smokers (Table 3).

**Table 3. Crude number of deaths (%) according to respiratory causes in the pooled cohort***.

|  | All pulmonary | Lung cancer | COPD | Pneumonia | Asthma | Interstitial pneumonia | Other pulmonary |
|---|---|---|---|---|---|---|---|
| Age, years |  |  |  |  |  |  |  |
| 15–29 | 90 (2) | 57 (2) | 21 (1) | 6 (1) | 2 (1) | 2 (1) | 2 (1) |
| 30–44 | 593 (10) | 376 (15) | 140 (8) | 44 (3) | 6 (5) | 12 (8) | 15 (9) |
| 45–59 | 2052 (34) | 1007 (41) | 601 (35) | 295 (22) | 39 (33) | 50 (34) | 60 (34) |
| ≥60 | 3214 (54) | 1002 (41) | 955 (56) | 1003 (74) | 72 (61) | 83 (56) | 99 (56) |
| Sex |  |  |  |  |  |  |  |
| Male | 4971 (84) | 2138 (88) | 1454 (85) | 1018 (76) | 87 (73) | 126 (86) | 148 (84) |
| Female | 978 (16) | 304 (12) | 263 (15) | 330 (24) | 32 (27) | 21 (14) | 28 (16) |
| Highest attained education |  |  |  |  |  |  |  |
| Compulsory education (<11 years) | 2656 (45) | 1022 (42) | 815 (47) | 642 (48) | 60 (50) | 52 (35) | 65 (37) |
| Medium level (11–13 years) | 2861 (48) | 1237 (51) | 797 (46) | 600 (45) | 56 (47) | 75 (51) | 96 (55) |
| University level (>13 years) | 432 (7) | 183 (7) | 105 (6) | 106 (8) | 3 (3) | 20 (14) | 15 (9) |
| Smoking status |  |  |  |  |  |  |  |
| Never | 781 (13) | 109 (4) | 119 (7) | 442 (33) | 16 (13) | 41 (28) | 54 (31) |
| Previous | 1378 (23) | 434 (18) | 366 (21) | 424 (31) | 34 (29) | 69 (47) | 51 (29) |
| Current | 3790 (64) | 1899 (78) | 1232 (72) | 482 (36) | 69 (58) | 37 (25) | 71 (40) |
| No. of cigarettes per day |  |  |  |  |  |  |  |
| 0–9 | 995 (21) | 324 (15) | 299 (21) | 291 (37) | 30 (35) | 25 (26) | 26 (24) |
| 10–19 | 2283 (49) | 1092 (51) | 715 (49) | 337 (43) | 39 (46) | 46 (48) | 54 (49) |
| ≥20 | 1395 (30) | 732 (34) | 442 (30) | 150 (19) | 16 (19) | 25 (26) | 30 (27) |
| Occupational exposure gas/dust |  |  |  |  |  |  |  |
| Yes | 2921 (49) | 1290 (53) | 926 (54) | 487 (36) | 56 (47) | 81 (55) | 81 (46) |
| No | 3028 (51) | 1152 (47) | 791 (46) | 861 (64) | 63 (53) | 66 (45) | 95 (54) |
| Cohort study |  |  |  |  |  |  |  |
| Oslo 72 | 1262 (21) | 427 (17) | 333 (19) | 400 (30) | 46 (39) | 23 (16) | 33 (19) |
| Hordaland 85 | 154 (3) | 62 (3) | 47 (3) | 36 (3) | 3 (3) | 2 81) | 4 (2) |
| Støvlunge 88–90 | 4313 (72) | 1854 (76) | 1263 (74) | 887 (66) | 66 (55) | 115 (78) | 128 (73) |
| Oslo/Hordaland 98–99 | 220 (4) | 99 (4) | 74 (4) | 25 (2) | 4 (3) | 7 (5) | 11 (6) |
| Total | 5949 (100) | 2442 (100) | 1717 (100) | 1348 (100) | 119 (100) | 147 (100) | 176 (100) |

*Some numbers do not add to a total of 100 percent because of rounding

The hazard of death due to pulmonary disease was higher in men compared with women, in those with only compulsory education compared with university education and in smokers versus never smokers (Table 4). Scores on breathlessness, cough and phlegm and attacks of breathlessness and ever wheezing were strongly associated with increased hazard of respiratory mortality. However, breathlessness showed a stronger association with pulmonary mortality than cough and phlegm did.

The pattern of associations was similar in subgroups, except that associations for phlegm and cough were weaker and non-significant in women and never-smokers, as well as for attacks of breathlessness/wheeze in men and never-smokers (Table 5).

## Lung cancer

The crude MR of lung cancer was 100 per 100,000 person-years. Ex-smokers had fourfold and current smokers tenfold increased hazard of lung cancer death relative to never smokers. Breathlessness, cough and phlegm were also associated with an increased risk of lung cancer death, with an increased risk occurring from a score of 1 to 2 on both symptom scales (Table 4). Attacks of breathlessness and ever wheeze did not increase the risk for lung cancer deaths. Identical trends for risks of respiratory symptoms for lung cancer deaths were observed in stratified analyses of men, women, never smokers and in those without self-reported cardio-pulmonary diseases (S4 Table).

## Chronic obstructive pulmonary disease

The MR was 70 for COPD per 100,000 person-years. The pattern of adjusted HR for COPD with regard to sex and attained education was almost identical to the HR for lung cancer, but with lower HRs with regard to smoking habits (Table 4). The HR with increasing breathless-ness on effort was higher for COPD than for other pulmonary causes of death.

The risk trends of respiratory symptoms for COPD deaths were similar for men, women, never smokers and for those without cardiopulmonary diagnosis (S5 Table).

## Pneumonia

The MR of pneumonia was 55 per 100,000 person-years. The MR of pneumonia increased markedly for those aged ≥60 years at baseline compared with younger individuals (Table 3). There was no clear association between smoking habits and death due to pneumonia. There was a dose-response relationship between breathlessness score and pneumonia deaths, while this was not so obvious for cough and phlegm score (Table 4).

Associations between symptom burden and pneumonia deaths were not prominent in women. Attacks of breathlessness and wheeze score was negatively associated with deaths due to pneumonia in men (S6 Table).

## Rare causes

Asthma was a rare cause of death with MR of 5 per 100,000 person-years, which was only 7% that of COPD deaths. The risk of asthma deaths showed the same pattern as COPD with regard to respiratory symptoms. However, higher attacks of breathless and wheeze score increased the risk of asthma mortality more than any other pulmonary causes of death (Table 4).

Interstitial pneumonia had a MR of 6 per 100,000 person-years. The risk pattern was similar for sex and education as for other pulmonary causes of deaths. The risk was lower in ex-smok-ers than in never smokers. An increased hazard of death from interstitial pneumonia was observed in those with breathlessness on effort (Table 4).

**Table 4. Hazard ratios (HR) for death with 95% confidence intervals and p-values according to pulmonary cause of death, multivariable proportional hazards regression analysis (n = 103,881).**

| | All pulmonary | | Lung cancer | | COPD | | Pneumonia incl. influenza | | Asthma | | Interstitial pneumonia | | Other pulmonary | |
|---|---|---|---|---|---|---|---|---|---|---|---|---|---|---|
| | HR | 95%CI | HR | 95%CI | HR | 95%CI | HR | 95%CI | HR | 95%CI | HR | 95%CI | HR | 95%CI |
| **Sex** | | | | | | | | | | | | | | |
| Male | 1 | | 1 | | 1 | | 1 | | 1 | | 1 | | 1 | |
| Female | 0.69*** | [0.64,0.75] | 0.60*** | [0.52,0.69] | 0.61*** | [0.52,0.72] | 0.88 | [0.75,1.03] | 0.8 | [0.48,1.31] | 0.60* | [0.36,0.99] | 0.55* | [0.34,0.89] |
| **Highest attained education** | | | | | | | | | | | | | | |
| Compulsory education (<11 years) | 1 | | 1 | | 1 | | 1 | | 1 | | 1 | | 1 | |
| Medium level (11–13 years) | 0.48*** | [0.46,0.51] | 0.55*** | [0.50,0.60] | 0.47*** | [0.43,0.52] | 0.38*** | [0.34,0.42] | 0.54** | [0.37,0.79] | 0.59*** | [0.41,0.84] | 0.58*** | [0.42,0.80] |
| University level (>13 years) | 0.20*** | [0.18,0.22] | 0.25*** | [0.21,0.29] | 0.19*** | [0.16,0.24] | 0.13*** | [0.11,0.16] | 0.09*** | [0.03,0.28] | 0.36*** | [0.21,0.62] | 0.19*** | [0.11,0.34] |
| **Smoking** | | | | | | | | | | | | | | |
| Never | 1 | | 1 | | 1 | | 1 | | 1 | | 1 | | 1 | |
| Previous | 1.88*** | [1.72,2.06] | 4.16*** | [3.37,5.14] | 2.92*** | [2.37,3.60] | 1.24** | [1.08,1.42] | 2.18* | [1.18,4.03] | 1.92** | [1.28,2.87] | 1.05 | [0.71,1.56] |
| Current | 2.76*** | [2.55,2.99] | 10.57*** | [8.69,12.86] | 4.63*** | [3.82,5.62] | 0.76** | [0.66,0.87] | 1.63 | [0.92,2.88] | 0.59* | [0.37,0.94] | 0.9 | [0.62,1.32] |
| **Occupational exposure to air pollution** | | | | | | | | | | | | | | |
| Yes | 1 | | 1 | | 1 | | 1 | | 1 | | 1 | | 1 | |
| None | 1.33*** | [1.26,1.41] | 1.21*** | [1.11,1.32] | 1.38*** | [1.24,1.53] | 1.64*** | [1.45,1.85] | 1.63* | [1.08,2.44] | 0.80 | [0.56,1.15] | 1.22 | [0.88,1.68] |
| **Breathless on effort, score** | | | | | | | | | | | | | | |
| 0 | 1 | | 1 | | 1 | | 1 | | 1 | | 1 | | 1 | |
| 1 | 1.44*** | [1.33,1.55] | 1.23*** | [1.09,1.39] | 1.96*** | [1.71,2.24] | 1.27* | [1.06,1.52] | 1.21 | [0.65,2.24] | 1.6 | [0.95,2.70] | 2.22*** | [1.45,3.40] |
| 2 | 2.21*** | [2.03,2.40] | 1.84*** | [1.61,2.10] | 3.47*** | [3.01,3.99] | 1.66*** | [1.34,2.06] | 2.78*** | [1.61,4.80] | 1.56 | [0.79,3.09] | 1.03 | [0.47,2.26] |
| 3 | 1.64*** | [1.51,1.79] | 2.63*** | [2.14,3.23] | 6.44*** | [5.37,7.73] | 2.99*** | [2.20,4.07] | 6.34*** | [3.51,11.44] | 4.84*** | [2.29,10.25] | 4.61*** | [2.14,9.93] |
| 4 | 1.74*** | [1.56,1.93] | 1.69** | [1.17,2.45] | 6.70*** | [5.27,8.51] | 3.05*** | [1.88,4.95] | 5.33*** | [2.51,11.31] | 3.64** | [1.04,12.81] | 8.72*** | [3.38,22.51] |
| **Cough and phlegm, score** | | | | | | | | | | | | | | |
| 0 | 1 | | 1 | | 1 | | 1 | | 1 | | 1 | | 1 | |
| 1 | 2.09*** | [1.87,2.34] | 1.24*** | [1.11,1.38] | 1.67*** | [1.46,1.91] | 1.08 | [0.93,1.26] | 0.96 | [0.53,1.74] | 1.25 | [0.80,1.93] | 1.07 | [0.71,1.61] |
| 2 | 2.16*** | [1.90,2.44] | 1.64*** | [1.44,1.87] | 1.97*** | [1.67,2.32] | 1.50** | [1.22,1.84] | 1.03 | [0.50,2.09] | 1.32 | [0.69,2.56] | 1.78* | [1.08,2.94] |
| 3 | | | 1.70*** | [1.45,1.99] | 2.34*** | [1.95,2.80] | 1.37* | [1.04,1.81] | 2.02* | [1.06,3.87] | 0.64 | [0.20,2.07] | 0.28 | [0.07,1.16] |
| 4 | | | 1.72*** | [1.43,2.07] | 3.10*** | [2.58,3.73] | 1.36 | [0.97,1.90] | 3.34*** | [1.81,6.15] | 2.34* | [1.06,5.19] | 0.94 | [0.36,2.43] |
| 5 | | | 2.06*** | [1.70,2.51] | 2.96*** | [2.43,3.62] | 1.43 | [0.97,2.11] | 1.47 | [0.69,3.16] | 1.82 | [0.67,4.94] | 1.26 | [0.49,3.21] |
| **Attacks of breathlessness and wheeze, score** | | | | | | | | | | | | | | |
| 0 | 1 | | 1 | | 1 | | 1 | | 1 | | 1 | | 1 | |
| 1 | 1.06 | [0.99,1.14] | 1.05 | [0.95,1.16] | 1.33** | [1.18,1.50] | 0.81** | [0.69,0.95] | 2.94*** | [1.74,4.98] | 0.71 | [0.43,1.18] | 0.60* | [0.37,0.97] |
| 2 | 1.16** | [1.06,1.27] | 0.98 | [0.85,1.14] | 1.64*** | [1.41,1.90] | 0.64*** | [0.49,0.83] | 6.48*** | [3.65,11.52] | 1.05 | [0.55,1.99] | 0.96 | [0.53,1.75] |

* p<0.05
** p<0.01
*** p<0.001

**Table 5. Hazard ratios for death with 95% confidence intervals and p-values, all pulmonary causes of death, in subgroups of the total population.** Multivariable proportional hazards regression analysis.

| | Men | | Women | | Never smokers | | Without cardiopulmonary disease | |
|---|---|---|---|---|---|---|---|---|
| | Hazard ratio | 95%CI | Hazard ratio | 95%CI | Hazard ratio | 95%CI | Hazard ratio | 95%CI |
| **Highest attained education** | | | | | | | | |
| **Compulsory education (<11 years)** | 1 | | 1 | | 1 | | 1 | |
| **Medium level (11–13 years)** | 0.50*** | [0.47,0.53] | 0.42*** | [0.37,0.48] | 0.35*** | [0.30,0.41] | 0.56*** | [0.51,0.61] |
| **University level (>13 years)** | 0.21*** | [0.19,0.24] | 0.12*** | [0.09,0.16] | 0.13*** | [0.10,0.17] | 0.22*** | [0.18,0.26] |
| **Smoking** | | | | | | | | |
| **Never** | 1 | | 1 | | | | 1 | |
| **Previous** | 2.32*** | [2.08,2.58] | 1.05 | [0.83,1.32] | | | 2.03*** | [1.75,2.36] |
| **Current** | 3.41*** | [3.09,3.77] | 1.67*** | [1.44,1.93] | | | 2.61*** | [2.27,3.00] |
| **Occupational exposure to air pollution** | | | | | | | | |
| **Yes** | 1 | | 1 | | 1 | | 1 | |
| **No** | 1.37*** | [1.30,1.46] | 1.00 | [0.84,1.19] | 1.58*** | [1.33,1.88] | 1.30*** | [1.18,1.42] |
| **Breathless on effort, score** | | | | | | | | |
| 0 | 1 | | 1 | | 1 | | 1 | |
| 1 | 1.52*** | [1.40,1.66] | 1.00 | [0.82,1.23] | 1.65*** | [1.28,2.11] | 1.47*** | [1.31,1.66] |
| 2 | 2.35*** | [2.14,2.58] | 1.54*** | [1.25,1.90] | 2.16*** | [1.64,2.84] | 2.39*** | [2.12,2.69] |
| 3 | 4.06*** | [3.57,4.62] | 2.79*** | [2.12,3.67] | 3.29*** | [2.16,5.01] | 4.05*** | [3.50,4.67] |
| 4 | 3.96*** | [3.28,4.80] | 2.45*** | [1.68,3.56] | 4.27*** | [2.54,7.17] | 3.61*** | [2.96,4.41] |
| **Cough and phlegm, score** | | | | | | | | |
| 0 | 1 | | 1 | | 1 | | 1 | |
| 1 | 1.31*** | [1.21,1.42] | 1.11 | [0.93,1.32] | 0.91 | [0.73,1.12] | 1.21** | [1.07,1.36] |
| 2 | 1.74*** | [1.58,1.92] | 1.19 | [0.95,1.50] | 1.24 | [0.88,1.75] | 1.51*** | [1.31,1.74] |
| 3 | 1.87*** | [1.67,2.09] | 1.17 | [0.88,1.54] | 1.43 | [0.96,2.13] | 1.58*** | [1.35,1.85] |
| 4 | 2.14*** | [1.89,2.42] | 1.85*** | [1.40,2.45] | 1.3 | [0.77,2.19] | 2.12*** | [1.82,2.48] |
| 5 | 2.32*** | [2.02,2.65] | 1.51* | [1.10,2.07] | 2.60*** | [1.59,4.24] | 1.96*** | [1.66,2.31] |
| **Attacks of breathlessness and wheeze, score** | | | | | | | | |
| 0 | 1 | | 1 | | 1 | | 1 | |
| 1 | 1.02 | [0.95,1.10] | 1.30** | [1.10,1.54] | 1.15 | [0.91,1.44] | 1.12* | [1.01,1.24] |
| 2 | 1.08 | [0.98,1.20] | 1.70*** | [1.36,2.11] | 1.29 | [0.93,1.79] | 1.06 | [0.94,1.20] |
| **Sex** | | | | | | | | |
| **Male** | | | | | 1 | | 1 | |
| **Female** | | | | | 1.22* | [1.01,1.47] | 0.68*** | [0.59,0.77] |
| N | 81510 | | 22371 | | 34916 | | 26723 | |

* p<0.05

** p<0.01

*** p<0.001

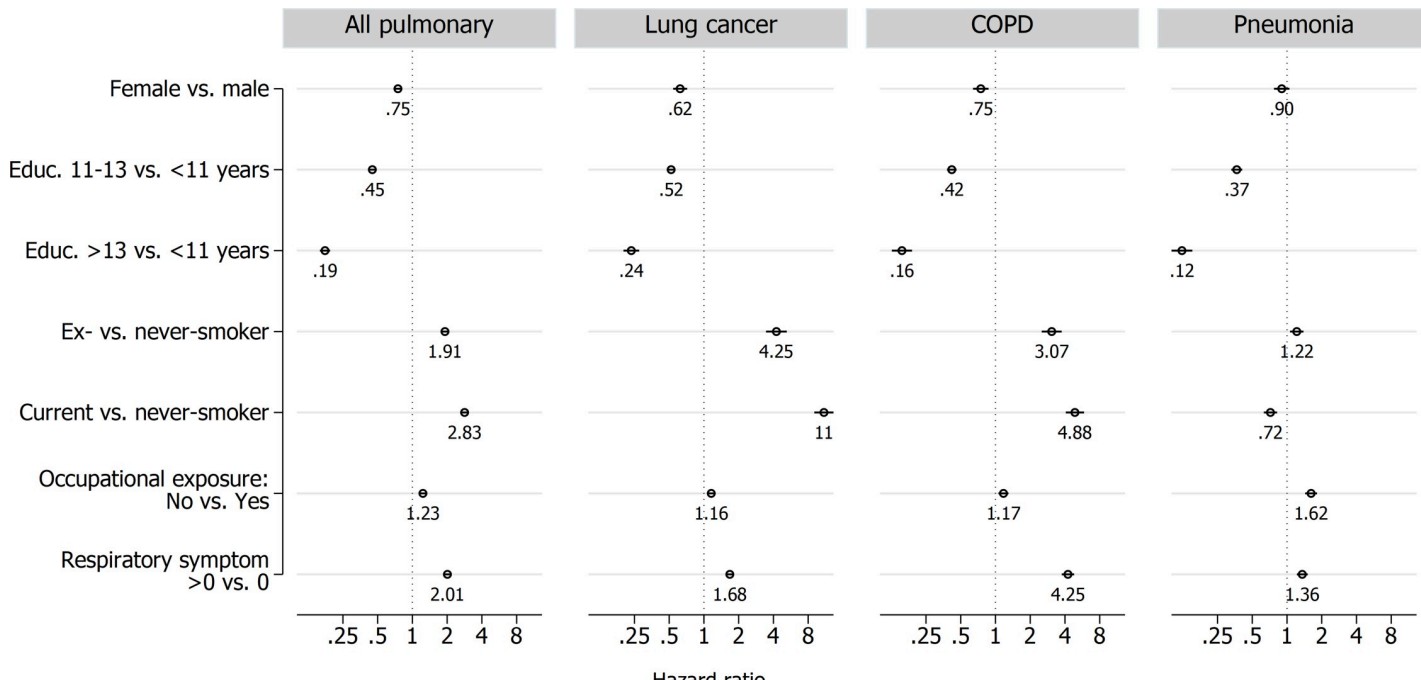

**Fig 1. Associations with cause-specific mortality.** Multivariable hazard ratios with 95% confidence intervals for any respiratory symptom (0 = no, 1 = yes), sex, education, smoking and occupational exposure to air pollution according to all pulmonary deaths, lung cancer, COPD or pneumonia. All subjects in pooled sample with available covariates (n = 107,136).

### Occupational exposure to air pollution

There was a slightly increased crude rate of pulmonary death in those with occupational exposure to gas or dust. However, no increased risk was evident in multivariable models. It was rather the opposite trend with a higher risk of deaths in those without occupational exposure to dust and gases compared with those with occupational exposure, except for death from interstitial pneumonia (Table 4).

### Any respiratory symptom

The presence of any respiratory symptom increased the HR (95%CI) for pulmonary death to 2.0 (1.9–2.1), lung cancer death to 1.7 (1.5–1.8), COPD death to 4.3 (3.8–4.8) and pneumonia to 1.4 (1.2–1.5) after adjustment for sex, education, smoking and occupational exposure (Fig 1).

### Discussion

This study has shown that respiratory symptoms were associated with respiratory deaths due to lung cancer, COPD, pneumonia and asthma. The scores of breathlessness, cough and phlegm had a dose-response relationship for specific causes of pulmonary deaths. These associations were stronger for deaths due to lung cancer and COPD than for pneumonia.

The burden of respiratory symptoms increased the risk of death due to lung disease, independent of subset of symptoms, although the subsets of respiratory symptoms were associated with different mortality risks. Breathlessness on effort was associated with a higher risk than cough and phlegm.

This study comprises a very large pooled cohort with a very long follow-up period, enabling us to present new data on cause-specific deaths. This is a strength of the present study and would not be possible or meaningful in smaller cohorts and explains why there is little previous information on the association between respiratory symptoms and the less common causes of respiratory deaths.

The study supports and extends the previous findings in one of the four cohorts with 30 years of follow up [12], which showed a strong association of respiratory symptoms with deaths due to obstructive lung disease and a weak association with pneumonia. We are aware of few other comparable studies with sub-classified respiratory death as outcomes. The Busselton study investigated the association between respiratory symptoms and respiratory deaths, except lung cancer which was grouped with cancers, with 20–26 years of follow-up. The study dichotomized the respiratory symptoms, whereas the present study had graded responses. The Busselton study included forced expiratory volume in 1 second ($FEV_1$) and several cardiovascular risk factors in the analysis of about 4,300 subjects, in contrast to the present study with >100,000 subjects that included occupational exposure and education in the analysis, but no spirometry variable. The present study was far larger and had longer observation time than that study, which enabled the analysis of subgroups of respiratory deaths, such as lung cancer, COPD, pneumonia, and less common respiratory diseases as causes of death.

In a sample of the population of Copenhagen, Denmark, chronic respiratory symptoms were associated with deaths in individuals with normal spirometry and without known airway disease [8]. In another follow up study, dyspnoea was associated with all-cause mortality after adjustment for lung function, but specific pulmonary mortality was not analysed [23]. In a large primary care study, breathlessness was an early marker of chronic respiratory and cardiac disease, and it was associated with all-cause mortality, death from COPD as well as ischemic heart disease [24].

We did not find an association between symptoms of attacks of breathlessness/wheeze and death from lung cancer. In contrast, a recent study reported a possible association between asthma symptom control and the incidence of lung cancer [25].

Our findings are robust given that the pattern of associations between respiratory symptoms and cause-specific mortality was similar in strata of men and subjects without cardiopulmonary diseases, and the total pooled sample. In stratified analysis of women only and never-smokers, the effects were smaller or disappeared, in particular for cough and phlegm.

Strengths of the present study include a large population-based cohort study with randomly selected individuals with high response rate of the target population and with a long and complete follow-up. The outcome of respiratory-specific deaths is highly relevant in a clinical setting. Furthermore, the respiratory symptoms were self-reported and not biased by observers [4]. The items on respiratory symptoms were identical in all these sub-cohorts, and they were administered by postal questionnaires. All studies had the same primary investigator.

In the present population-based study, it is unlikely that we have missed individuals with substantial symptoms, but more likely that we have included more individuals with mild symptoms. Nevertheless, individuals with mild symptoms will only dilute an association and reduce the risk estimates. There was a clear dose-response relationship between respiratory symptoms and outcomes of death like lung cancer and COPD.

Some limitations of the study should be noted. In this large, pooled sample, we had access to only a limited number of common/harmonized covariates. It is possible that covariates such as cardiovascular risk factors, body mass index, genetic factors, more detailed history of allergy or infections may, or other variables may be confounders in the relationship between respiratory symptoms and the specific respiratory deaths. During such a long observation time, covariates may also change over time, however, this study only had available covariates at baseline.

Spirometry would have been a useful supplement to subjective perceptions of respiratory symptoms, but that was only available for a fraction of the participants in the pooled sample. However, spirometry is more resource-consuming than assessment of simple respiratory symptoms in clinical practice.

The largest sub-cohort included mostly only men, hence the pooled sample had relatively few women and may represent a bias, although we adjusted for sex in the analysis. In a previous report, however, we have shown a similar pattern of association between respiratory symptoms and all-cause mortality for men and women in stratified analyses [6].

As in any survey, there is a risk of non-participation bias; however, it is hard to delineate how this would impact the association between respiratory symptoms and the studied outcome. The response rate in the surveys in this study ranged 67–89%, which is higher than what would probably be achievable today, as survey response rates to epidemiological studies have decreased over the past decades [26]. However, response rates depend on the topic and length of the questionnaire, target population, and whether incentives are used [26–28].

Another potential limitation is that ICD codes are reported by medical doctors, and there will be misclassifications. A validity study of the European short list of respiratory death certificate and autopsy showed a very high agreement for lung cancer, intermediate for COPD but only fairly for pneumonia [29]. This type of misclassification is likely to be non-differential and would bias toward the null [30, 31], and cannot explain our positive results.

Pneumonia is a difficult outcome based on cause of death coding, and there might be different predisposing conditions. It is possible that deaths coded as pneumonia as underlying cause of death could reflect unrevealed chronic lung diseases, or be related to other covariates than those that were available in the pooled sample in these cohorts.

Occupational exposure to air pollution was not a significant predictor of deaths for lung cancer, COPD and pneumonia in the pooled cohorts in this study. On the contrary, we observed a reduced risk for lung cancer mortality as well as COPD mortality in those with occupational exposure to dust and gas. This can be due to the healthy worker effect, i.e. a selection of relatively healthier individuals into occupations with dusty exposures. Paradoxically, in 1985 we found that a positive answer to the same question on occupational exposure was associated with an increased risk of respiratory symptoms [32]. Further explanations for this reduced risk could be information biases with greater reduction in smoking, improved socio-economic status or less obesity in the follow up of the occupational exposure subset of the population compared with the rest of the population, or possibly better health care or other health-promoting factors for those with occupational exposure.

Smoking might be a confounder of the association between respiratory symptoms and mortality outcomes. Although we controlled for smoking before inclusion in the study, the prevalence of smoking has changed during follow up. In a follow-up of men in a Norwegian workforce, 32% had quit smoking, and only 16% had started smoking over a period of 8 years [33]. In addition, the total proportion of daily smokers in Norway decreased from 42% to 9% during the follow up period [34]. As a considerable proportion of the persons who smoked and reported symptoms might have reduced their symptoms after quitting [35], our analysis probably underestimates rather than overestimates the relation between symptoms and cause-specific pulmonary mortality.

Our study does not elucidate why having respiratory symptoms increases the risk for mortality from lung cancer, COPD and pneumonia, or why self-reports of such symptoms at any time have such long lasting effect. This study was an epidemiological study and not designed to investigate physiological mechanisms, therefore we can only speculate on such mechanisms. Furthermore, we do not have data that can confirm why for instance, dyspnea was a powerful predictor of COPD death, but a weaker predictor of lung cancer and pneumonia death.

It possible that the dyspnea domain captures both deconditioning by pulmonary, cardiovascular disease or other comorbidities, as well as anxiety-related conditions which may be associated with gradually developing COPD and death from COPD, and that this is less relevant for lung cancer or pneumonia, which have more rapid onset. In contrast, cough and phlegm might be more related to smoking habits and it is possible that there may be residual confounding even after adjustment for smoking. In conclusion, scores of respiratory symptoms were associated with long-term deaths due to COPD and lung cancer. Health personnel should be encouraged to ask patients about their symptom burden, as this may be a simple and cost-effective way to map increased mortality risk also among those without known illness. Awareness of even common respiratory symptoms is important in detecting deadly underlying diseases.

## Supporting information

**S1 Checklist.**
(DOCX)

**S1 Table. Questions (Q) on respiratory symptoms and scores.**
(PDF)

**S2 Table. Descriptive statistics for participants at different stages according to response to questionnaire.**
(PDF)

**S3 Table. Prevalence of baseline symptoms and distribution of symptoms according to principal causes of respiratory death.**
(PDF)

**S4 Table. Hazard ratios (HR) with 95% confidence intervals and p-values for lung cancer death according to subgroup, multivariable proportional hazards regression analysis.**
(PDF)

**S5 Table. Hazard ratios (HR) with 95% confidence intervals and p-values for COPD death according to subgroup, multivariable proportional hazards regression analysis.**
(PDF)

**S6 Table. Hazard ratios (HR) with 95% confidence intervals and p-values for death from pneumonia according to subgroup, multivariable proportional hazards regression analysis.**
(PDF)

## Author Contributions

**Conceptualization:** Knut Stavem, Ane Johannessen, Rune Nielsen, Amund Gulsvik.

**Data curation:** Knut Stavem, Amund Gulsvik.

**Formal analysis:** Knut Stavem, Ane Johannessen, Amund Gulsvik.

**Investigation:** Knut Stavem, Amund Gulsvik.

**Methodology:** Knut Stavem, Ane Johannessen, Rune Nielsen, Amund Gulsvik.

**Project administration:** Amund Gulsvik.

**Resources:** Amund Gulsvik.

**Software:** Knut Stavem.

**Supervision:** Rune Nielsen, Amund Gulsvik.

**Validation:** Knut Stavem.

**Visualization:** Knut Stavem, Ane Johannessen, Rune Nielsen, Amund Gulsvik.

**Writing – original draft:** Knut Stavem, Amund Gulsvik.

**Writing – review & editing:** Knut Stavem, Ane Johannessen, Rune Nielsen, Amund Gulsvik.

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
