## [Decision Letter · Decision Letter 0]

12 Jul 2021

PONE-D-21-08054

Respiratory symptoms and respiratory deaths:

a multi-cohort study over 45 years

PLOS ONE

Dear Dr. Stavem,

Thank you for submitting your manuscript to PLOS ONE. After careful consideration, we feel that it has merit but does not fully meet PLOS ONE’s publication criteria as it currently stands. Therefore, we invite you to submit a revised version of the manuscript that addresses the points raised during the review process.

At this point we would request you to carefully go through the reviewers' comments and address them satisfactorily. There are several issues raised by the reviewers which you may address through approprite revisions or inclusions in your manuscript. 

We look forward to receiving your revised manuscript.

Kind regards,

Koustubh Panda, M. Tech., Ph.D

Academic Editor

PLOS ONE

Journal Requirements:

"I have read the journal's policy and the authors of this manuscript have the following competing interests: RN reports grants and personal fees from AstraZeneca, grants from GlaxoSmithKline, grants from Boehringer Ingelheim, outside the submitted work. All other authors have declared that no competing interests exist."

Reviewers' comments:

Reviewer's Responses to Questions

**Comments to the Author**

1. Is the manuscript technically sound, and do the data support the conclusions?

Reviewer #1: No

Reviewer #2: Yes

Reviewer #3: Yes

Reviewer #4: Yes

2. Has the statistical analysis been performed appropriately and rigorously? 

Reviewer #1: No

Reviewer #2: Yes

Reviewer #3: I Don't Know

Reviewer #4: Yes

3. Have the authors made all data underlying the findings in their manuscript fully available?

Reviewer #1: No

Reviewer #2: No

Reviewer #3: No

Reviewer #4: No

4. Is the manuscript presented in an intelligible fashion and written in standard English?

Reviewer #1: Yes

Reviewer #2: Yes

Reviewer #3: Yes

Reviewer #4: Yes

5. Review Comments to the Author

Reviewer #1: The study by Stavem et al. attempts to establish an association between respiratory symptoms and death from respiratory causes (including pneumonia, lung cancer, COPD, asthma, and other pulmonary diseases) in subjects over 45 years of age. However, the study has several shortcomings that seriously undermine the overall premises on which the conclusions have been drawn or the title has been framed for the reported study.

Major concerns:

The overall study design is not conforming to its title or the conclusions drawn. What is of particular concern is that -

(i) presence of possible or associated comorbidities in the subjects have not been considered which makes it questionable on whether the recorded deaths apparently from ‘respiratory causes’ can be attributed exclusively to respiratory symptoms

(ii) The study is significantly gender biased (84% male vs. 16% female) with a mean baseline age of 46.8 years which is extremely skewed for a cohort representing subjects above the age of 45 years. In fact, the basis for inclusion of 12% subjects in the age group of 15-44 years in a study aiming to examine respiratory symptoms related mortality in subjects above the age of 45 years is not only unclear but this contributes to the above aberrant effect on the age statistics as well as unnecessarily introduces complications in the data analysis because the factors governing respiratory diseases among such young group of subjects and the relatively aged group of 45 plus that the study tends to represent are extremely different.

(iii) Moreover, BMI, vitamin D deficiency, genetic factors, history of allergy and infections etc. can also play a determining role in the development of respiratory symptoms and associated mortality. Unfortunately, there is no consideration or mention of these factors in the study

(iv) The data used in the study have been collected between the year 1972-1999 and all subjects were born between 1902 to1973. There are thus grounds to suppose that the study methodologies and techniques used for assessment of the defined respiratory symptoms have not been consistent over such a long period

(v) The pathophysiological dynamics of various pulmonary diseases vary widely especially when parameters like cough and phlegm are used as ‘respiratory symptoms’. Thus, a more uniform and dependable approach like spirometric assessment of respiratory performance (eg. FEV1 and FVC) would have perhaps been more valuable for a study of this nature.

Minor concerns:

1. There have been several studies that have examined the relation between respiratory symptoms and mortality in specific age groups. The manuscript fails to critically portray these studies in sufficient detail and thereby enlighten how and why the present study is different from similar studies of the past (e.g. Busselton Health Study by Knuiman et.al.)

2. The authors express restrictions on data availability. The scope of such restrictions should have been defined in more detail as availability of all raw data related to studies of the present nature is somewhat essential.

In summary, the present study lacks sound design and critical supportive data to conclusively establish a relation between respiratory symptoms and mortality especially in subjects over the age of 45. Overall, the study fails to contribute any substantive knowledge or information that is new or valuable to our present understanding of the factors underlying respiratory diseases and related mortality.

Reviewer #2: In this cohort study, the authors evaluated the association of respiratory symptoms with mortality due to respiratory causes in Norway by leveraging population based health outcome data (National Cause of Death Registry) and cross sectional symptom surveys in 1972, 1985, 1988, and 1998. Although the authors did not make all data underlying their manuscript fully available, their rationale and a pathway for access to this data to confirm findings is reasonable. I’ve included my comments below

- Infection related pneumonia is not generally considered a chronic respiratory condition as mentioned in the beginning of the introduction. If you are referring to interstitial/histiopathologic pneumonia (such as UIP, EIP, etc)., please clarify this in the introduction.

- What was the temporal association between symptoms and deaths due to pneumonia? Given the observation of increased breathlessness and death due to pneumonia, and the presumption that all deaths did not occur shortly after the survey, is it possible there was another intermediate factor that increased the risk of death due to pneumonia (such as undiagnosed chronic lung disease, bronchiectasis etc)?

- Please address the following limitations: Male biased cohort, limitations associated with survey data regarding selection bias.

Reviewer #3: First, a caveat. On the surface, the methods seem OK, but my expertise is dyspnea mechanisms and measurement, not epidemiologic stats. I am not equipped to make a serious critique of epidemiologic statistical methods in this study.

This analysis utilizes data collected in 4 previous cross sectional studies conducted between 1972 and 1990 in Norway utilizing a Norwegian adaptation of the MRC respiratory questionnaire in 100,000 people. The authors examine the relationship between respiratory symptoms and cause of death. Of the eligible subjects (born between 1902 and 1973), 44,000 had died by the end of 2016 when data collection was terminated. This is a pretty heterogeneous group – subjects would have been 43 to 114 years of age at the study termination, and there would have been a wide variation in the time between survey and death. The main finding of the paper is that subjects who reported respiratory symptoms during a population survey are more likely to die from respiratory disease than subjects who do not report respiratory symptoms. The sample was then subcategorized by respiratory disease (COPD, lung cancer, pneumonia …). The strongest association was between COPD as cause of death and reported breathlessness (increasing breathlessness score associated with increasing hazard ratio - a breathlessness score of 4 was associated with a 7-fold risk of COPD death). These findings are unsurprising to those familiar with the literature.

The information in this paper may be useful to some, but it would benefit from a less narrow view. I am in agreement with the authors’ conclusion that “Health personnel should be encouraged to ask patients about their symptom burden, as this may be a simple and cost-effective way to map increased mortality risk also among those without known illness. Awareness of even common respiratory symptoms is important in detecting deadly underlying diseases.” In fact, I have been supporting this idea for years. There is, however, substantial published information supporting this idea (much of which has not been cited by the authors). I’m not sure the present work, in its current form, adds much. Perhaps the authors could consider enhancing its usefulness.

1) The paper has a narrow focus on a particular epidemiologic question, and lacks broader context. There are several sources for broader information on dyspnea and cough. You could fit this together with your observations – why for instance, would dyspnea be such a powerful predictor of COPD death, but a much weaker predictor of lung cancer and pneumonia death? Why does it predict pneumonia death at all? Pneumonia is an acute disease. These people were not surveyed as part of a medical encounter, so they would be unlikely to have pneumonia symptoms at the time of survey. Is the connection indirect, through COPD? One might also consider the measurement circumstances and timing – what was the gap between survey and death? What was the age at death, etc. Also one might consider the physiologic mechanisms causing cough and breathlessness – how are these related to disease process, and does this inform the interpretation of the link between symptom and cause of death? The questionnaire used (a revision of the MRC) is quite old (consistent with the date of the data collection) – how does that limit the study? Do you think more modern instruments would improve sensitivity or specificity? Why? Or why not?

2) Can you argue the advantages to the clinician or to public health agencies of routinely assessing and documenting respiratory symptoms? It is not at all surprising or new that respiratory symptoms are associated with respiratory deaths. But are there advantages not elucidated in the paper? Do symptoms predict mortality better than other common indicators in your database? For example look at the key paper by Nishimura (Chest 2002) showing that dyspnea is a better predictor of COPD mortality than FEV1. Do symptoms predict mortality before other common indicators? (time from first report of symptoms to death may be within your database, but is not reported). Other important papers relating respiratory symptoms to mortality are missing from your references and discussion (eg, Abidov NEJM 2005, Santos PLOS One 2016).

3) Dyspnea usually indicates some problem in oxygen delivery and utilization, or problems with CO2 elimination – thus it is associated with many cardiovascular and metabolic diseases in addition to those classified as respiratory disease. Any consideration of dyspnea as a predictor of these other diseases is absent from your work, although these causes of deaths are probably in your database.

4) Information on prevalence of the various scores of the symptoms would be useful – both overall in the sample, and subdivided as in tables S3, S4, S5. By this I mean, for instance, how many people reported a score of 3 on dyspnea, etc.

5) A curious detail: Why do you suppose that a dyspnea score of 4 was associated with fewer lung cancer deaths than a score of 3?

Reviewer #4: The research study reported respiratory symptoms and death from respiratory causes. In summary, 14% of the examined population died from respiratory causes. Of these, 41% died from lung cancer, 29% from COPD, 23% from pneumonia, 2% from asthma, 2% from interstitial lung disease and 3% from other pulmonary diseases.

Minor revisions:

1- Line 191: State the proportion of men who died.

2- Title and Abstract: Clarify that 45 years refers to a span of time, not the participant's ages.

6. PLOS authors have the option to publish the peer review history of their article (what does this mean?). If published, this will include your full peer review and any attached files.

Reviewer #1: No

Reviewer #2: No

Reviewer #3: No

Reviewer #4: No

---

## [Author Response · Author response to Decision Letter 0]

28 Sep 2021

Rebuttal letter with comments to editor and reviewers has been uploaded as a file.

---

## [Decision Letter · Decision Letter 1]

10 Nov 2021

Respiratory symptoms and respiratory deaths:

a multi-cohort study with 45 years observation time

PONE-D-21-08054R1

Dear Dr. Stavem,

We’re pleased to inform you that following a review of your revised submission your manuscript has been judged scientifically suitable for publication and will be formally accepted for publication once it meets all outstanding technical requirements.

Kind regards,

Koustubh Panda, M. Tech., Ph.D

Academic Editor

PLOS ONE

---

## [Editor Report · Acceptance letter]

12 Nov 2021

PONE-D-21-08054R1 

Respiratory symptoms and respiratory deaths: a multi-cohort study with 45 years observation time 

Dear Dr. Stavem:

I'm pleased to inform you that your manuscript has been deemed suitable for publication in PLOS ONE. Congratulations! Your manuscript is now with our production department. 

Kind regards, 

on behalf of

Professor Koustubh Panda 

Academic Editor

PLOS ONE